# CROSS-GRANULARITY VIDEO LLM FOR LONG VIDEO UNDERSTANDING

## ABSTRACT

Video Large Language Models (video LLMs) have demonstrated remarkable capabilities in video understanding tasks, such as video question answering and temporal localization. However, understanding long videos still remains a significant challenge. Existing video LLMs adopt uni-granularity tokens for long videos, failing to simultaneously understand both high-level semantics and low-level visual details in videos. To tackle the problem, we propose **CrossVLLM**, a video LLM framework with adaptive modules of different granularities to collaborate with each other for long video understanding, which not only retains the capabilities of high-level video semantics understanding, but also strengthens the fine-grained understanding abilities. Specifically, we propose the coarse-to-fine grounding and fine-to-coarse reflection strategies for long video understanding. In the coarse-to-fine grounding strategy, video LLM with a coarse-grained module first locates the key video segments from the long video by tackling massive frames of the long video with fewer per-frame tokens. And then video LLM adapted with the fine-grained module further analyzes the key video segments with more per-frame tokens so that it can understand fine-grained information. In case the video LLM locates the wrong key video segments, during the inference stage, our designed fine-to-coarse reflection strategy instructs the fine-grained module to reflect the effectiveness of the locating result and decide whether to return to the coarse-to-fine grounding strategy with reflection feedback. Additionally, during the training stage, the coarse-to-fine grounding strategy is optimized with our proposed cross-granularity reinforcement learning strategy to further improve grounding efficiency. Extensive experiments for long video question answering and temporal video grounding tasks demonstrate that our proposed CrossVLLM framework can significantly improve the Video Large Language Model for long video understanding.

## 1 INTRODUCTION

Recently, Video Large Language Models (video LLMs) have made significant progress in video understanding tasks such as video question answering and temporal localization (Zhang et al., 2023). By leveraging techniques like modality alignment and visual instruction tuning, several models (Huang et al., 2024; Ren et al., 2024; Li et al., 2023) have been developed to improve temporal video representation learning and comprehension.

Despite the success of video LLMs, understanding long videos still remains a significant challenge (Wu et al., 2024; Zhou et al., 2025; Tan et al., 2025). Compared to short videos, long videos involve much more frames and thus require much more tokens that may exceed the token length limits of the LLMs (Weng et al., 2024; Shen et al., 2025). Existing video LLMs for long videos generally adopt uni-granularity tokens to represent the videos and then adopt the token pruning methods or frame sampling methods to reduce the token number (Song et al., 2024; Huang et al., 2024; Bai et al., 2023). However, the token pruning methods may suffer from losing visual details, and the frame sampling methods will even lose high-level temporal semantic context.

To address the problems, as shown in Figure 1, different from existing works that adopt uni-granularity tokens and then reduce tokens, we propose **CrossVLLM**, a novel video LLM framework for long video understanding with adaptive modules of different granularities to reflectively

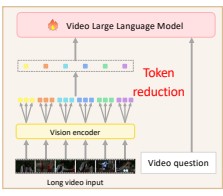 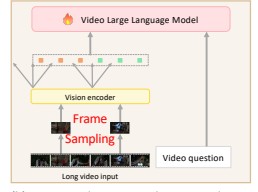 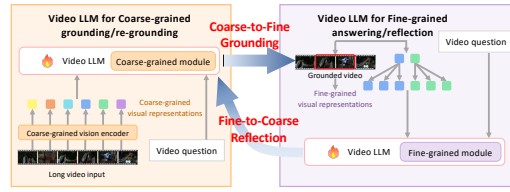

(a) Existing Video LLMs with uni-granularity video tokens through token reduction

(b) Existing Video LLMs with uni-granularity video tokens through frame sampling

(c) Our Cross-granularity video LLM framework with cross-granularity video tokens for long video understanding

Figure 1: Comparison between existing works and our cross-granularity video LLM framework. Figure 1(a) denotes token reduction for long video understanding, which limits the video LLMs for detailed video perception. Figure 1(b) represents the frame sampling method, which would ignore massive intermediate question-related frames when analyzing long videos.

collaborate with each other. The proposed **CrossVLLM** works similarly to the way that humans try to answer a question for a given long video, where we first skim through the video to roughly understand the global semantics of the video, and then watch the segments relevant to the question in a fine-grained manner. Specifically, the proposed CrossVLLM framework consists of i) a video LLM with a coarse-grained module to process more visual frames under fewer per-frame visual tokens, tackling massive frames of the long video, and ii) a video LLM with a fine-grained module processing fewer frames through more per-frame visual tokens. With the cross-granularity modules, we design coarse-to-fine grounding and fine-to-coarse reflection strategies for cross-granularity long video understanding. During the coarse-to-fine grounding strategies, we first utilize the video LLM with the coarse-grained module to locate the key video segments from the long video according to the textual input. And then instruct the video LLM adapted with the fine-grained module to further analyze the key video segments with more per-frame tokens so that it can understand fine-grained visual information. In case the video LLM locates the wrong key video segments, during the inference stage of our cross-granularity method, we design the fine-to-coarse strategy, prompting the video LLM with the fine-grained module to reflect the effectiveness of the locating result, and decide whether to return information to the coarse-to-fine grounding strategy with prior reflection. In addition, during the training stage, we design our cross-granularity reinforcement learning strategy to further optimize the coarse-to-fine grounding strategy with grounding feedback. Extensive experiments show that our CrossVLLM framework is able to significantly outperform existing video LLMs in long video question answering and temporal video grounding tasks.

To summarize, we make the following contributions:

- To the best of our knowledge, the proposed CrossVLLM is the first attempt at cross-granularity video LLM for long video understanding.
- We propose coarse-to-fine grounding and fine-to-coarse reflection strategies for the CrossVLLM, where the coarse-to-fine grounding strategy is further optimized with our designed cross-granularity reinforcement learning strategy.
- Extensive experiments show that our CrossVLLM outperforms state-of-the-art baseline methods in long video question answering and temporal grounding tasks, indicating its superiority for high-level semantics and low-level visual details in long video understanding.

## 2 RELATED WORK

### 2.1 VIDEO LARGE LANGUAGE MODEL

With the rapid development of Large Language Models (Touvron et al., 2023), significant research has been devoted to enabling LLMs for temporal visual information understanding (Zhang et al., 2023; Li et al., 2023). These Large Language Models capable of processing video input, could be collectively referred to as Video Large Language Models. Many video LLMs, such as Video-LLaMA (Zhang et al., 2023; Cheng et al., 2024), VideoChat (Li et al., 2023), and Video-LLaVA (Lin

et al., 2024) share similar methods with image LLMs (Bavishi et al., 2023; Bai et al., 2023) through inputting entire image patches or massive visual tokens (with more than 200 tokens per frame) to the transformer architecture (Waswani et al., 2017) of LLMs. These methods, however, all have critical challenges when dealing with long video understanding tasks due to exceeding the maximum token limitations for visual representation or only processing a small number of sampled video frames in the video, resulting in the loss of keyframe capture.

## 2.2 Video LLMs for Long Video Understanding

To address the critical challenges of long video understanding, video LLMs such as LLaMA-VID (Li et al., 2024b), MovieChat (Song et al., 2024), and LongVU (Shen et al., 2025) have been developed for long video understanding through compressing each video frame into one or a few visual tokens so that the language model is able to handle all the visual representations from the entire video input (Shu et al., 2025). However, since these methods are pre-trained from a short video understanding model and reduced visual representation features of each video frame, they are unable to analyze detailed visual information for long videos.

On the other hand, some methods such as VideoTree (Wang et al., 2025) and VideoAgent (Wang et al., 2024b) attempted to explore two-stage strategies in the format of first selecting keyframes for the next-stage detailed perception (Tang et al., 2025). However, these methods primarily employ training-free approaches or only involve basic model fine-tuning, failing to optimize the overall strategy. They overlook the reflection refinement and specific training for strategies when the model incorrectly selects keyframes, which would compromise the accuracy of long video analysis.

## 3 The Proposed CrossVLLM Framework

In this section, we will introduce our CrossVLLM framework. As shown in Figure 2, our CrossVLLM framework includes video LLMs with a coarse-grained module and a fine-grained module, collaborating through our designed coarse-to-fine grounding and fine-to-coarse reflection strategies. In the coarse-to-fine grounding strategy, video LLM processes a large number of frames from the long video using fewer visual tokens per frame, allowing the coarse-grained module to efficiently identify key video segments. The key segment is then passed to the fine-grained video LLM, which analyzes it in greater detail using more tokens per frame to capture fine-grained visual information. In the fine-to-coarse reflection strategy, in case the coarse-grained video LLM locates the wrong key video segments, the fine-grained video LLM will reflect the effectiveness of the grounding result and determine whether to go back to the coarse-to-fine grounding strategy for re-grounding correct segments.

### 3.1 Coarse-to-Fine Grounding

The coarse-to-fine grounding includes a coarse-grained grounding model and a fine-grained answering model, which are our pre-finetuned coarse-grained video LLM and fine-grained video LLM, respectively.

**Coarse-grained Grounding.** Given a long video input $v \in R^{T \times H \times W \times C}$ with $T$ frames, the coarse-grained video LLM would uniformly sample $N$ frames, represented as $\widetilde{v} \in R^{N \times H \times W \times C}$, and process these frames through the vision transformer (ViT) independently:

$$\{v_i^{cls}, v_i^1, v_i^2, ..., v_i^{patch}\} = ViT(\widetilde{v}_i), i = 1, 2, ..., N, \tag{1}$$

where $patch$ represents the number of patches of the ViT. Utilizing the global feature $v_i^{cls}$ as the feature for the i-th frame. The video LLM then applies a projection layer to map the ViT features into the feature space of the LLM:

$$
\begin{aligned}
z_i &= f(v_i^{cls}), i = 1, 2, ..., N, \\
Z &= \{z_i\} \in R^{N \times L \times d},
\end{aligned}
\tag{2}
$$

where $Z$ is the input sequence LLM able to understand, $d$ is the dimension of LLM's hidden space, and $N \times L$ is the total number of visual tokens for the input video. Noted that for the video LLM with different settings, the parameters $N$ (*number of sampled frames*) and $L$ (*number of tokens per*

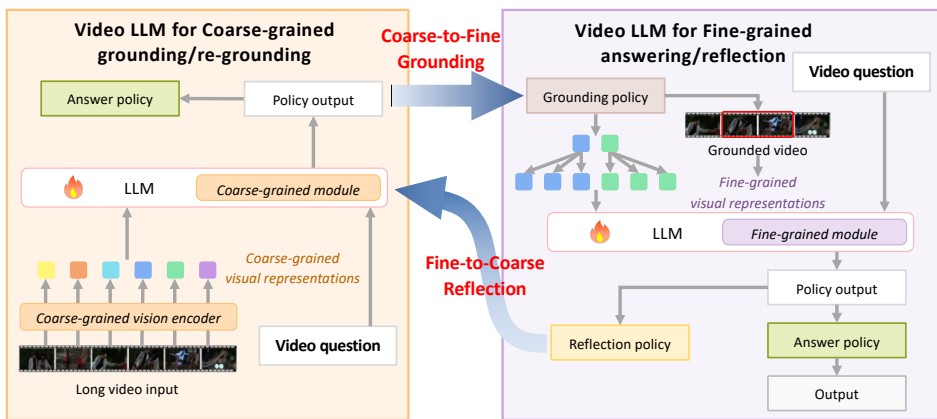

Figure 2: Our CrossVLLM framework, including a video LLM with a coarse-grained module to process more visual frames, and a video LLM with a fine-grained module processing more per-frame visual tokens of fewer frames. In our designed coarse-to-fine grounding strategy, the coarse-grained video LLM would localize the question-related short video segment and send it to the fine-grained video LLM for detailed perception. In case the coarse-grained video LLM locates the wrong video segments, during our fine-to-coarse reflection strategy, the fine-grained video LLM will reflect the effectiveness of the grounding result and determine whether to go back to the coarse-to-fine grounding strategy for re-grounding with prior feedback.

*frame*) from the above equations may vary[1]. Since we require the coarse-grained video LLM to have the global understanding capability and temporal boundary perception capability on the input long videos. To achieve that, compared to the default setting of the original video LLM, we increase the *number of sampled frames* $N$ to process more more video frames and gain a more comprehensive global understanding of the input long video while decreasing the *number of tokens per frame* $L$ at a lower level, to ensure that $N \times L$ does not exceed the LLM processing limit for visual tokens.

After the processing of visual information, we introduce our **coarse-grained grounding prompt** $P_c$ in the textual format (details are provided in the supplement), which is designed to stimulate the video LLM to temporally localize the most relevant short video segments to the textual question input and send the temporal grounding results. To enable the LLM aware of the original length of the input video before grounding prediction, we insert the statement *'The length of the original video is $t_{total}$ seconds'* in the prompt $P_c$, where $t_{total}$ is the total duration of the original long video. The coarse-grained prompt $P_c$ and the original video question $q$ are first combined and transformed into the textual embedding list $[w_1, w_2, ..., w_M] = Tokenizer(P_c, q)$, where $w_i \in R^d$ is the embedding of the word token and $M$ is the word number. Then, the video feature sequence will be inserted to form the input content for the video LLM:

$$input = [Z, w_1, w_2, ..., w_M], \tag{3}$$

so that the video LLM can further encode the input embedding to understand the video, related question, and our coarse-grained prompt, and give the final response:

$$p_{grounding} = VidLLM_{corase-grained}(input),$$
$$p_{grounding} = \{t_{start}, t_{end}, answer\}, \tag{4}$$

where $p_{grounding}$ is the output policy prompted in the JSON format and contains the predicted start timestamp $t_{start}$ and end timestamp $t_{end}$ of the video segment that the video LLM judges is most relevant to the original video question. Since video LLM with the coarse-grained module may encounter relatively simple video questions for short videos, it is also prompted allowed to give policies with direct output $answer$ in simple video cases.

**Fine-grained Answering.** With the predicted start timestamp $t_{start}$ and end timestamp $t_{end}$, we obtain the video segment $v' \in R^{T' \times H \times W \times C}$ with $T' = T\frac{t_{end}-t_{start}}{t_{total}}$ frames. The fine-grained

---
[1]Normally LLMs have token limits. Therefore, the value of $N \times L$ cannot exceed a certain threshold.

video LLM would sample $N'$ frames represented as $\widetilde{v} \in R^{N' \times H \times W \times C}$ and process visual frames similar to the coarse-grained module:

$$\{v_i^{cls}, v_i^1, v_i^2, ..., v_i^{patch}\} = ViT(\widetilde{v}_i), i = 1, 2, ..., N',$$
$$z_i = f'(v_i^{cls}), i = 1, 2, ..., N', \tag{5}$$
$$Z = \{z_i\} \in R^{N' \times L' \times d}.$$

Since the fine-grained video understanding module is designed for video LLM to analyze the localized short video content related to the input query, which is relatively shorter, and its visual information is more worthy of in-depth extraction and understanding. Therefore, we increase the *number of tokens per frame $L$* compared to the default setting of the original video LLM, enabling the model to have fine-grained perception under visual frames from the relatively short video while decreasing the *number of sampled frames $N$* to keep $N \times L$ within the capabilities of LLM.

After the processing of visual information, we transform the original video question $q$ into a textual embedding list $[w_1, w_2, ..., w_M] = Tokenizer(q)$ to encode the input embedding for the short video segment understanding:

$$input = [Z, w_1, w_2, ..., w_M],$$
$$output = VidLLM_{fine-grained}(input), \tag{6}$$

and we obtain the detailed perception answer *output* responded by the video LLM adapted with the fine-grained module, which analyzes the key video segment with more per-frame visual tokens.

### 3.2 Fine-to-Coarse Reflection

The fine-to-coarse reflection includes a fine-grained reflection model and a coarse-grained re-grounding model, which are consistent with the fine-grained video LLM and coarse-grained video LLM of the last section. Considering it is possible for the coarse-grained video LLM to produce incorrect grounding results in a single inference, during our fine-to-coarse reflection strategy, in addition to generating normal responses under fine-grained visual representations, the video LLM with the fine-grained module is also prompted to provide reflective judgments on the validity of the received short video segments and decide whether it requires re-grounding the relevant video segments.

**Fine-grained Reflection.** The visual information processing during the fine-grained reflection is the same as that of the fine-grained answering module. Different from the textual processing of fine-grained answering, we design our **fine-to-coarse reflection prompt** $P_f$ in the textual format (details are provided in the appendix), which is able to instruct the video LLM to provide reflective judgments on the validity of the received grounded video segments. The fine-grained prompt $P_f$ and the original video question $q$ would be combined and transformed into the textual embedding list $[w_1, w_2, ..., w_M] = Tokenizer(P_c, q)$ for short video understanding and reflection:

$$input = [Z, w_1, w_2, ..., w_M],$$
$$p_{reflection} = VidLLM_{fine-grained}(input), \tag{7}$$
$$p_{reflection} = \{reflection, answer\},$$

where the output policy in the JSON format contains two aspects, which are $reflection$ and $answer$. The $reflection$ denotes the textual reason that the grounded short video is not suitable for the original question and requires another attempt for related segment grounding, and it would be returned to the coarse-grained re-grounding module. If the video LLM judges the localized video to be suitable for question answering, the $reflection$ would be empty and $answer$ would be the final output to the video question.

**Coarse-grained Re-grounding.** If the judgement of the fine-to-coarse reflection indicates that the grounded short video is not related to the video question, this $reflection$ information of the temporalization would be transformed into the format of '$t_{start}$ *to* $t_{end}$ *is not the suitable grounding segments because...*' and it will be returned to the coarse-grained video LLM.

The visual information processing during the coarse-grained re-grounding is the same as that of the coarse-grained grounding module. During the textual processing, we will first combine the reflection

and coarse-grained prompt to form a new coarse-grained prompt $P_c = (P_c, reflection)$ with the last time grounding feedback. And the updated coarse-grained prompt $P_c$ is transformed similarly $[Z, w_1, w_2, ..., w_M] = Tokenizer(P_c, q)$ compared to the coarse-grained grounding module for re-grounding:

$$
\begin{aligned}
input &= [Z, w_1, w_2, ..., w_M], \\
p_{grounding} &= VidLLM_{coarse-grained}(input), \\
p_{grounding} &= \{t'_{start}, t'_{end}, answer\},
\end{aligned}
\tag{8}
$$

where the output policy $p_{grounding}$ contains the new start timestamps and end timestamps that the video LLM predicts are relevant to the original video question and different from the last time of coarse-grained grounding.

For a single video question in our complete CrossVLLM framework, the fine-grained reflection module would replace the fine-grained answering module, and the coarse-grained re-grounding module could be considered as a coarse-grained grounding module with at least one feedback prior. Therefore, two cross-granularity modules could be called multiple times, until the video LLM with the fine-grained module responds with positive feedback on the short video segment, or the video LLM with the coarse-grained module only responds with direct output policy without temporal grounding result. If the calls of two modules exceed a certain threshold limit, the calls will also terminate and select the answer response from the last round of video LLM as output.

## 4 MODULE TRAINING

In this section, we will introduce how we optimize the modules of video LLM with different granularities. Our training method includes the fine-tuning of the coarse-grained module, the fine-tuning of the fine-grained module, and most importantly, the cross-granularity reinforcement learning for modules with different granularities through the coarse-to-fine strategy.

### 4.1 COARSE-GRAINED AND FINE-GRAINED FINE-TUNING

Formally, given a question $q$ through tokenization and related video $v$ through visual processing for video LLM, the supervised fine-tuning loss can be defined as the cross-entropy loss as follows:

$$
L_{CE}(\widehat{y}(q,v), y) = - \sum_{(v,q,y) \in D} y \log(\widehat{y}(q,v)),
\tag{9}
$$

where $y$ is the ground truth answer in the token sequence and $D$ is the dataset.

**Coarse-grained Module Training.** To effectively localize short segments from the original video that are relevant to the question, we collect the training datasets of temporal video grounding datasets including Charades-STA (Gao et al., 2017), ActivityNet-Captions (Krishna et al., 2017), and VTG-IT (Guo et al., 2025) to finetune the coarse-grained module, which will receive the global visual representations transformed from the complete video content and predict the most relevant short video segments to the question input.

**Fine-grained Module Training.** The fine-grained module is designed for video LLM to analyze the localized short video content related to the input query, which is relatively shorter, and its visual information is more worthy of in-depth extraction and understanding. Therefore, we select the training subset of video data with a duration of less than one minute from several VideoQA datasets, including ActivityNet-QA (Yu et al., 2019), Ego-QA (Grauman et al., 2022), and Next-QA (Xiao et al., 2021), to fine-tune another fine-grained module to complete the fine-grained understanding and reasoning tasks for short videos.

### 4.2 CROSS-GRANULARITY REINFORCEMENT LEARNING

Since the task of providing temporal localization policies for video-related questions still has some differences from the retrieval input textual query, which the coarse-grained video LLM is trained with, it is difficult to use in-context learning alone to ensure the quality of policies. To address this issue, we propose a reinforcement learning-based cross-granularity training method to optimize

cross-granularity modules of video LLM continuously for processing long videos, which enables the video LLM to improve the temporal grounding quality of generated policies, and adapt to the key video segment for fine-grained understanding.

**Training Strategy.** Given a question $q$ about the input long video $v$, we prompt the coarse-grained video LLM to generate several different temporal localization policies $\{p_j\}$ of question-relevant short video segments from the original long video. These policies would inform the fine-grained module to process the grounded short videos in more detailed visual representations and give the final answers. It could be inferred that a correct policy that more accurately localizes the temporal video segments related to the video question would be more helpful for the fine-grained video LLM. Consequently, the loss computed by the video LLM tends to be smaller. Meanwhile, negative policies that localize irrelevant video segments would mislead the video LLM, resulting in incorrect responses and an increase in loss. The details about our reinforcement learning strategy are provided in Algorithm 1 in the Appendix.

Therefore, we are able to provide feedback to the coarse-grained video LLM with the losses computed by the fine-grained video LLM for cross-granularity training. Inspired by the Reinforcement Learning from Human Feedback (RLHF) (Bai et al., 2022; Christiano et al., 2017), we apply Direct Preference Optimization (DPO) (Rafailov et al., 2024) to train the coarse-grained video LLM to generate more accurate policies. The DPO method directly optimizes the Video Large Language Model without explicit rewarding models and formulates the policy objective as:

$$L_{DPO}(\pi_\theta; \pi_{ref})$$
$$= -E_{(q,v,p_w,p_l)\sim D}[\log \sigma(\beta \log \frac{\pi_\theta(p_w|q)}{\pi_{ref}(p_w|q)} - \beta \log \frac{\pi_\theta(p_l|q)}{\pi_{ref}(p_l|q)})], \qquad (10)$$

where $p_w$ represents the positive with the smaller loss that accurately localizes the short video segment related to the video question, and $p_l$ denotes the negative policy with the larger loss. $\pi_\theta$ represents our coarse-grained video LLM to be trained in this stage, while $\pi_{ref}$ is a reference model also initialized with coarse-grained video LLM but remains frozen. $\sigma$ is the sigmoid function and $\beta$ is a controlling parameter.

Our cross-granularity reinforcement learning alternates between direct preference optimization for the coarse-grained video LLM and SFT for the fine-grained video LLM, which optimizes with the loss function in Equation 9 and 10. After the cross-granularity training, the coarse-grained video LLM is able to provide refined policies with more accurate localization segments, and the fine-grained video LLM would also adapt the localization of short videos from the coarse-grained video LLM. For training datasets, we select the subset from the NeXT-QA and ActiveNet-QA datasets with videos that exceed 2 minutes.

## 5 EXPERIMENTS

**Implementations.** We conduct our experiments on long video understanding and temporal video grounding datasets. We select VideoMME, Lvbench and MLVU for assessing the long video understanding ability (Fu et al., 2024; Wang et al., 2024a; Zhou et al., 2025). For the temporal video grounding task, we utilize the test set of ActivityNet Captions and Charades-STA. Our baselines include long-video video LLMs and fine-grained video LLMs for long video understanding, and baselines for temporal video grounding include state-of-the-art temporal perception video LLMs. We implement our fine-tuning and reinforcement learning method based on the SWIFT framework. We utilize LLaVA-Next-Video(7B) (Zhang et al., 2024b) as the backbone for both our coarse-grained and fine-grained video LLM with linear scale factor=2. More details are provided in the Appendix A.1.

### 5.1 EXPERIMENTS ON LONG VIDEO UNDERSTANDING

Based on the long video understanding benchmarks, we evaluate the capabilities of existing video LLMs in long video understanding tasks. As the results shown in Table 1, we can draw the following conclusion: (i) compared to existing video LLMs, our CrossVLLM achieves the overall best performance, and we even utilize small-scale base models (LLaVA-NeXT-Video-7B) outperform larger-scale identical base model (LLaVA-NeXT-Video-34B), demonstrating the effectiveness of

Table 1: Performance comparison of state-of-the-art video LLMs with our CrossVLLM methods on long video understanding benchmarks. The best average performance is in **bold** and the second is underlined.

| Models | Size | VideoMME | | | | Lvbench | MLVU |
| --- | --- | --- | --- | --- | --- | --- | --- |
| | | Short | Medium | Long | Overall | | |
| Duration(min) | | ≤2 | 4∼15 | 30∼60 | 1∼60 | 30∼140 | 3∼120 |
| Video-LLaVA (Lin et al., 2024) | 7B | 46.1 | 40.7 | 38.1 | 41.6 | 21.6 | 47.3 |
| Chat-UniVi (Jin et al., 2024) | 7B | 51.2 | 44.6 | 41.8 | 45.9 | 25.3 | 52.6 |
| ShareGPT4Video (Chen et al., 2024) | 8B | 53.6 | 39.3 | 37.9 | 43.6 | 21.8 | 46.4 |
| VideoChat2 (Li et al., 2024a) | 7B | 52.8 | 39.4 | 39.2 | 43.8 | 23.7 | 47.9 |
| LongVA (Zhang et al., 2024a) | 7B | 61.6 | 53.6 | 47.6 | 54.3 | 31.7 | 56.3 |
| Video-RAG with LLaVA-NeXT (Luo et al., 2024) | 7B | 56.6 | 47.4 | 46.0 | 50.0 | 30.2 | 53.5 |
| LongVU (Shen et al., 2025) | 7B | 64.7 | 58.2 | 59.5 | 60.9 | 38.3 | 65.4 |
| Video-XL (Shu et al., 2025) | 7B | 67.4 | 60.7 | 54.9 | 61.0 | 37.7 | 64.9 |
| VITA 1.5 (Fu et al., 2025) | 7B | 67.0 | 54.2 | 47.1 | 56.1 | 32.1 | 60.2 |
| LLaVA-NeXT-Video (Zhang et al., 2024b) | 34B | 65.1 | 52.2 | 47.2 | 54.9 | 32.2 | 61.6 |
| Ours-CrossVLLM | 7B | **70.9** | **64.4** | **61.9** | **65.7** | **41.8** | **70.4** |

our method, and (ii) under VideoMME evaluation, as video duration increases, the performance of all the methods declines. Nevertheless, thanks to our coarse-to-fine grounding and fine-to-coarse reflection strategies, our CrossVLLM performance on long videos still surpasses all evaluated models, including state-of-the-art long video understanding models.

## 5.2 EXPERIMENTS ON TEMPORAL VIDEO GROUNDING

We evaluate the capabilities of existing video LLMs in temporal video grounding tasks shown in Table 2. Given a video and a textual query, the models are required to identify the start and end timestamps of the video segment corresponding to the query in the video. During our cross-granularity inference strategy, our coarse-to-fine strategy would localize the related short video segment. The fine-grained video LLM would receive an additional description of *'the short video is sampled from <start time> to <end time> of the original video'* and would be informed to further localize the temporal grounding segment under more detailed visual representations from the short video. We can see that our CrossVLLM outperforms video LLMs trained through temporal perception on both metrics of recall and mean IoU.

Table 2: Performance comparison of state-of-the-art temporal perception video LLMs with our CrossVLLM methods on temporal video grounding tasks. The best average performance is in **bold** and the second is underlined.

| Models | ActivityNet Captions | | | | Charades-STA | | | |
| --- | --- | --- | --- | --- | --- | --- | --- | --- |
| | R@0.3 | R@0.5 | R@0.7 | mIoU | R@0.3 | R@0.5 | R@0.7 | mIoU |
| ChatVTG (Qu et al., 2024) | 40.7 | 22.5 | 9.4 | 27.2 | 52.6 | 33.0 | 15.9 | 34.9 |
| VtimeLLM (Huang et al., 2024) | 44.0 | 27.8 | 14.3 | 30.4 | 51.0 | 27.5 | 11.4 | 31.2 |
| TimeChat (Ren et al., 2024) | - | - | - | - | - | 32.2 | 13.4 | - |
| Momentor (Qian et al., 2024) | 42.9 | 23.0 | 12.4 | 29.3 | 42.6 | 26.6 | 11.6 | 28.5 |
| VTG-LLM (Guo et al., 2025) | - | - | - | - | 52.0 | 33.8 | 15.7 | - |
| NumPro (Wu et al., 2025) | 45.5 | 30.8 | 18.4 | 33.6 | 60.7 | 36.8 | 15.9 | 38.5 |
| BTDP (Deng et al., 2025) | 50.6 | 30.6 | 17.5 | 36.6 | 58.3 | 40.0 | 20.9 | 39.1 |
| Ours-CrossVLLM | **53.9** | **41.7** | **23.1** | **39.8** | **63.5** | **44.9** | **21.0** | **42.6** |

## 5.3 ABLATION STUDY

In this section, we provide detailed ablation analyses of our cross-granularity strategies through experiments on our models to evaluate the effectiveness of our designed modules. The results are shown in Table 3.

### 5.3.1 RESULT ABOUT FINE-TUNING AND REINFORCEMENT LEARNING

Based on the results of Row 1,3,4 and 5 from Table 3, we can see the positive impact through both coarse-grained and fine-grained fine-tuning on video LLMs. The utilization of coarse-grained fine-tuning comprehensively improves the video LLMs in processing both temporal video grounding and long video understanding tasks. The fine-grained fine-tuning helps mainly in short video understanding and temporal video grounding. In addition, although only applying basic model fine-tuning does not significantly improve overall performance compared to applying our entire cross-granularity framework, they still occupy an important position in our method, as shown in Row 1 and 2, we can see that directly using not fine-tuned video LLMs for fine-to-coarse reflection and reinforcement learning would not bring performance improvement, encountering a certain degree of cold start problem.

Table 3: Ablation study of our cross-granularity strategies. The coarse-to-fine grounding strategy is activated through the entire experiment. The *Tune* in the line of *Coarse* and *Fine* respectively represent coarse-grained training and fine-grained training for video LLMs while *Freeze* represents image-text feature alignment only. *RL* represents the utilization of cross-granularity reinforcement learning. The best average performance is in **bold**.

| Row | Coarse | Fine | RL | Fine-to-Coarse Reflection | ActivityNet Captions | | | | VideoMME | | | |
|-----|--------|------|-----|-----------|-------|-------|-------|-------|-------|--------|------|---------|
| | | | | | R@0.3 | R@0.5 | R@0.7 | mIoU | Short | Medium | Long | Overall |
| 1 | Freeze | Freeze | ✗ | ✗ | 31.3 | 16.0 | 6.8 | 22.4 | 55.2 | 44.0 | 42.7 | 47.3 |
| 2 | Freeze | Freeze | ✓ | ✓ | 31.8 | 16.6 | 6.9 | 23.2 | 56.8 | 44.6 | 41.5 | 47.6 |
| 3 | Tune | Freeze | ✗ | ✗ | 42.3 | 25.8 | 11.6 | 29.4 | 58.2 | 45.4 | 43.7 | 49.1 |
| 4 | Freeze | Tune | ✗ | ✗ | 38.7 | 20.4 | 8.6 | 25.9 | 60.3 | 44.5 | 42.2 | 49.0 |
| 5 | Tune | Tune | ✗ | ✗ | 44.7 | 28.1 | 14.2 | 31.2 | 61.6 | 47.2 | 44.9 | 51.2 |
| 6 | Tune | Tune | ✓ | ✗ | 52.3 | 38.9 | 20.4 | 38.5 | **71.8** | 60.3 | 53.8 | 62.0 |
| 7 | Tune | Tune | ✗ | ✓ | 47.9 | 30.5 | 15.6 | 32.4 | 67.7 | 57.8 | 51.4 | 58.9 |
| 8 | Tune | Tune | ✓ | ✓ | **53.9** | **41.7** | **23.1** | **39.8** | 70.9 | **64.4** | **61.9** | **65.7** |

### 5.3.2 RESULT ABOUT CROSS-GRANULARITY REINFORCEMENT LEARNING

In Row 5, 7 vs Row 6, 8 from Table 3, we removed the cross-granularity reinforcement learning strategy and conducted evaluations on benchmarks. The results show a decrease in all the evaluation metrics. Compared to the results of other models shown in Table 1 and 2, our experiment without the reinforcement learning method only performs at an average level among baselines, indicating the necessity of the cross-granularity reinforcement learning strategy we proposed for training video LLMs to further improve grounding efficiency.

### 5.3.3 RESULT ABOUT FINE-TO-COARSE REFLECTION

During the inference stage, our fine-to-coarse reflection strategy prompts the fine-grained video LLM to reflect on the locating effectiveness and decide whether to return the response to the coarse-grained video LLM for re-grounding with reflection feedback. As shown in Row 7, 8 vs Row 5, 6, with the addition of fine-to-coarse reflection, the performance of our method has further improved, indicating our fine-to-coarse reflection provided by fine-grained video LLM contains useful information for coarse-grained video LLM to regenerate a suitable grounding policy.

## 6 CONCLUSION

In this paper, we propose CrossVLLM, a novel cross-granularity video LLM framework for long video understanding. Specifically, we design the coarse-to-fine grounding and fine-to-coarse reflection strategies utilizing adaptive modules of video LLM with different granularities to collaborate with each other for cross-granularity long video inference. We further propose a cross-granularity reinforcement learning strategy to optimize video LLM through training and inference when processing long videos. Extensive experiments demonstrate that CrossVLLM outperforms existing video LLMs in long video question answering and temporal video grounding tasks, indicating its superiority for processing both high-level semantics and low-level visual details in long video understanding.

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

# A APPENDIX

## A.1 IMPLEMENTATION DETAILS

We provide the implementation details of our work shown in Table 4. We implement our fine-tuning and reinforcement learning method based on the SWIFT framework. We utilize LLaVA-Next-Video(7B) (Zhang et al., 2024b) as the backbone for both our coarse-grained and fine-grained video LLM with linear scale factor=2. We set the *number of sampled frames N* and *number of tokens per frame L* to $(16, 12 \times 12 = 144)$ for fine-grained video LLM and $(144, 4 \times 4 = 16)$ for coarse-grained video LLM during training, and extended to $(32, 12 \times 12 = 144)$ and $(288, 4 \times 4 = 16)$ during inference enabled by linear scale factor. Before the fine-tuning of different granularity modules and reinforcement learning, we first train the visual adapter $f$ through the LLaVA (Liu et al., 2024) dataset for image-text feature alignment. Our first stage coarse-grained and fine-grained fine-tuning are conducted separately under one NVIDIA A100-40GB GPU. And our cross-granularity reinforcement learning for coarse-grained and fine-grained modules is conducted jointly under two A100-40GB GPUs.

Table 4: Implementation details

| Config | Coarse-grained finetuning | Fine-grained finetuning | Coarse-grained during RL | Fine-grained during RL |
|---|---|---|---|---|
| Video LLM | | LLaVA-Next-Video-7B | | |
| Optimizer | | AdamW | | |
| Epochs | 3 | 10 | | 2 |
| Warmup ratio | | 0.05 | | |
| Learning rate | 2e-5 | 1e-4 | | 2e-5 |
| Batch size | | 1 | | |
| Gradient accumulation steps | | 16 | | |
| linear scale factor | | 2 | | |
| Pooling Stride | 6 | 2 | 6 | 2 |
| Number of Tokens per Frame | $4 \times 4$ | $12 \times 12$ | $4 \times 4$ | $12 \times 12$ |
| Training cost | 87h | 41h | 350h | |
| number of policies per data $n$ | not required | | 4 | |

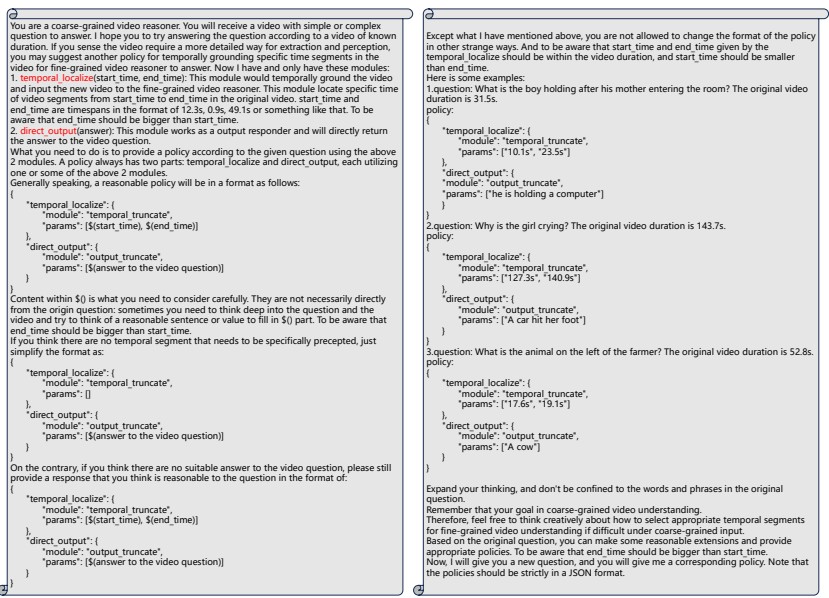

Figure 3: Prompt for coarse-grained video LLM

**Prompt for Video LLM.** We design coarse-grained and fine-grained prompts for the video LLM with the corresponding visual modules to generate policies from given questions based on the in-context learning capabilities of LLM. The complete prompts are presented in Figure 3 and 4. We first inform the video LLM with the basic information about the functional modules, and then provide the general template of the policy in the uniform JSON format.

```
You are a fine-grained video reasoner. You will receive a specific time segment from a video
with simple or complex question to answer. The temporal segment is grounded by a coarse-
grained video reasoner. I hope you to try judging whether this segment of the video related
to the question and answering the question. If you sense requiring a different way for
extraction and perception of another video segment, you may suggest another policy
reflected to corase-grained video reasoner. Now I have and only have these modules:
1. temporal_reflect(reflection): This works as a reflection module and will use reflection to
summarize whether the specific temporal segment is irrelevant to the video question, which
will return to the coarse-grained video reasoner for another round temporal grounding.
2. direct_output(answer): This module works as a output responder and will directly return
the answer to the video question.
What you need to do is to provide a policy according to the given question using the above
2 modules. A policy always has two parts: temporal_reflect and direct_output, each utilizing
one or some of the above 2 modules.
Generally speaking, a reasonable policy will be in a format as follows:
{
    "temporal_reflect": {
        "module": "reflect_truncate",
        "params": [$(reflection to the video segment)]
    },
    "direct_output": {
        "module": "output_truncate",
        "params": [$(answer to the video question)]
    }
}
Content within $() is what you need to consider carefully. They are not necessarily directly
from the origin question: sometimes you need to think deep into the question and the
video and try to think of a reasonable sentence or value to fill in $() part.
If you think he specific temporal segment is not irrelevant to the video question, just
simplify the format as:
{
    "temporal_reflect": {
        "module": "reflect_truncate",
        "params": []
    },
    "direct_output": {
        "module": "output_truncate",
        "params": [$(answer to the video question)]
    }
}
On the contrary, if you think there are no suitable answer to the video quesion, please still
provide a response that you think is reasonable to the question in the format of:
{
    "temporal_reflect": {
        "module": "reflect_truncate",
        "params": [$(start_time), $(end_time)]
    },
    "direct_output": {
        "module": "output_truncate",
        "params": [$(answer to the video question)]
    }
}
Except what I have mentioned above, you are not allowed to change the format of the policy
in other strange ways.
```

```
Here is some examples:
1.question: What is the boy holding after his mother entering the room? The segment is
sampled from 10.1s to 23.5s of the original video.
policy:
{
    "temporal_reflect": {
        "module": "reflect_truncate",
        "params": ["In the video from 10.1s to 23.5s, the mother can be seen but the boy
cannot be found"]
    },
    "direct_output": {
        "module": "output_truncate",
        "params": ["he may hold a basket"]
    }
}
2.question: Why is the girl crying? The segment is sampled from 127.3s to 140.9s of the
original video.
policy:
{
    "temporal_reflect": {
        "module": "reflect_truncate",
        "params": []
    },
    "direct_output": {
        "module": "output_truncate",
        "params": ["A car hit her foot"]
    }
}
3.question: What is the animal on the left of the farmer? The segment is sampled from 17.6s
to 19.1s of the original video.
policy:
{
    "temporal_reflect": {
        "module": "reflect_truncate",
        "params": []
    },
    "direct_output": {
        "module": "output_truncate",
        "params": ["A cow"]
    }
}
Expand your thinking, and don't be confined to the words and phrases in the original
question.
Remember that your goal in fine-grained video understanding.
Therefore, feel free to think creatively about whether and how to reflect the effectiveness of
the grounding segment.
Based on the original question, you can make some reasonable extensions and provide
appropriate policies.
Now, I will give you a new question, and you will give me a corresponding policy. Note that
the policies should be strictly in a JSON format.
```

Figure 4: Prompt for fine-grained video LLM

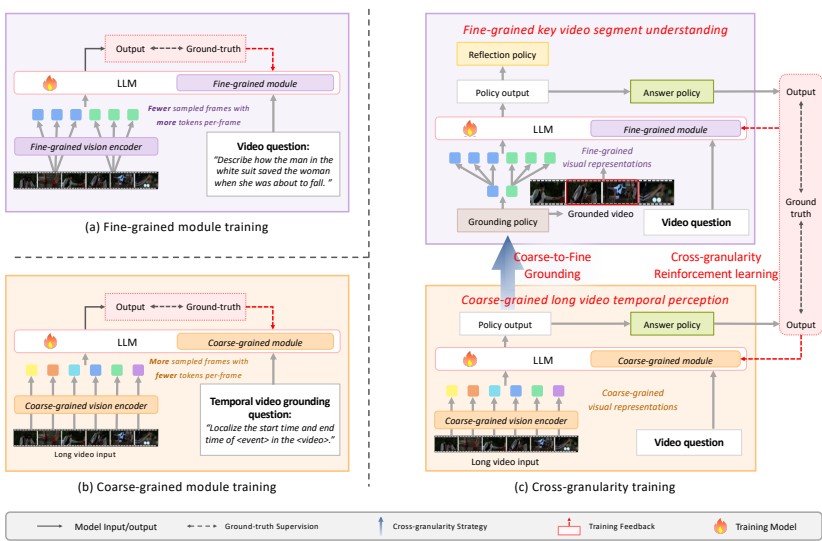

Figure 5: Detailed framework for the module training of CrossVLLM. The left part shows our fine-tuning strategies for adaptive modules of video LLM, including (a) a fine-grained module for detailed video understanding and (b) a coarse-grained module for temporal perception. The right part (c) demonstrates the training details of our cross-granularity reinforcement learning.

During the inference stage, both prompts would be applied entirely. During the training stage, only the coarse-grained prompt would be applied, and in order to stimulate the collaboration of modules with different granularities for video LLM, we replace the instruction of prompting coarse-grained video LLM to inference direct output, instructing video LLM to predict temporal grounding policy only. Therefore, the video LLM would learn how to generate the correct policy through the examples provided by the prompts and the reinforcement learning strategy mentioned in our main text.

**Module Training details.** For coarse-grained module training, we collect a 175k training dataset of temporal video grounding datasets including Charades-STA (Gao et al., 2017), ActivityNet-Captions (Krishna et al., 2017), and VTG-IT (Guo et al., 2025) to finetune the coarse-grained mod-

---

**Algorithm 1** Cross-granularity Training Strategy

---

1: **Input:** Coarse-grained module (coarse-grained video LLM) $C$, Temporal-Grounding module $G$, Coarse-grained prompt $P_c$, Fine-grained module (fine-grained video LLM) $F$, dataset $D = \{(v_i, q_i, y_i)\}_{i=1}^{N}$, training stepts $S$, gradient accumulate step $s$, numbers of policies per data $n$
2: **Freeze:** $F, G$, **Activate:** $C$
3: **for** t=1,...,S **do**
4:   initialize $\pi_\theta = C, \pi_{ref} = C$
5:   **for** m=1,...,s **do**
6:     $i \leftarrow ((t-1)s + m - 1)\%N + 1$
7:     Prepare data$(v_i, q_i, y_i)$ from $D$
8:     Generate policies $p_1, p_2, ...p_n = \pi_{ref}(P_c(q_i, v_i))$
9:     **for** j=1,...,n **do**
10:       $G$ **execution localization**: $v_j = G(v_i, p_j)$
11:       $F$ **forward propagation**: $\widehat{y} = F(q_i, v_j)$
12:       **Compute** $L_{CEj}$ in Equation(9)
13:     **end for**
14:     $p_w \leftarrow argmin_{\{p_j\}}L_{CE}$
15:     $p_l \leftarrow \{p_j, p \neq p_w\}$
16:     **Optimize** $\pi_\theta$ **with loss:**$L_{DPO}$ in Equation(10)
17:     Add$(v_i, q_i, y_i)$ to $CACHE$
18:   **end for**
19:   $C \leftarrow \pi_\theta$,freeze $C$, activate $F$
20:   **for** i=1,...,s **do**
21:     Prepare data $(v_i, q_i, y_i)$ from $CACHE$
22:     Generate single policy $p = Q(P_c(q_i, v_i))$
23:     $G$ **execution localization**: $v_p = G(v_i, p)$
24:     $F$ **forward propagation**: $\widehat{y} = F(q_i, v_p)$
25:     **Optimize** $F$ **with loss:**$L_{CE}$ in Equation( 9)
26:   **end for**
27:   clear $CACHE$, freeze $F$, activate $C$
28: **end for**

---

ule. For the fine-grained module training, we select a 21.5k training dataset of video data with a duration of less than one minute from several VideoQA datasets, including ActivityNet-QA (Yu et al., 2019), Ego-QA (Grauman et al., 2022), and Next-QA (Xiao et al., 2021), to fine-tune another fine-grained module to complete the fine-grained understanding and reasoning tasks for short videos. For cross-granularity reinforcement learning, we select a 7.5k training dataset from the NeXT-QA and ActiveNet-QA datasets with video data exceeding 2 minutes. We provide the detailed framework of module training shown in Figure 5 and reinforcement learning algorithm in Algorithm 1.

**Evaluation Metrics.** We conduct our experiments on long video understanding and temporal video grounding datasets. We select VideoMME, Lvbench and MLVU for assessing the long video understanding ability (Fu et al., 2024; Wang et al., 2024a; Zhou et al., 2025). Based on the multi-choice question, we calculate the matching accuracy between the output of the model and the correct answer. For the temporal video grounding task, we utilize the test set of ActivityNet Captions and Charades-STA. We evaluate the performance of different methods by calculating the Intersection over Union(IoU) between the time segments generated by the model and the time segments of the ground truth. We report the mean IoU(mIoU) and recall@1, IoU$\geq m$(R@m) as evaluation metrics.

**Baselines.** On long video understanding tasks, our baselines include long-video video LLMs LongVA (Zhang et al., 2024a), LongVU (Shen et al., 2025), Video-RAG (Luo et al., 2024), video-XL (Shu et al., 2025) and fine-grained video LLMs (Chen et al., 2024) such as Video-LLaVA (Lin et al., 2024), VideoChat2 (Li et al., 2024a), Chat-UniVi-V1.5 (Jin et al., 2024), ShareGPT4Video (Chen et al., 2024), LLaVA-NeXT-Video (Zhang et al., 2024b). Our baselines for temporal video grounding include temporal perception video LLMs such as VTimeLLM (Huang et al., 2024), TimeChat (Ren et al., 2024), Monmentor (Qian et al., 2024), VTG-LLM (Guo et al., 2025), NumPro (Wu et al., 2025), ChatVTG (Qu et al., 2024), and the LLM-based temporal grounding method BTDP (Deng et al., 2025).

## A.2 RESULT ABOUT HYPERPARAMETER SENSITIVITY

We modify the parameters on *number of sampled frames $N$* and *number of tokens per frame $L$*, to evaluate the hyperparameter sensitivity of the video LLM. The results are shown in Table 5 and 6, where Video LLMs are applied with coarse-grained and fine-grained training but without reinforcement learning. Based on Table 5, by increasing the processing granularity and decreasing the sampling frames for coarse-grained video LLM, our cross-granularity framework achieves higher performance on shorter video understanding while underperforming on long videos. These results can be attributed to the fact that the key content of short videos is not easily missed due to the low sampling rate, and the accuracy of understanding can be improved through higher visual processing granularity, while the key information of long videos is more likely to be missed by using a lower sampling rate, resulting in wrong answers. On the contrary, increasing the number of sampling frames would help ground key content in long videos, while decreasing the granularity would negatively affect detailed visual perception for shorter videos. Based on Table 6, the overall performance is optimal when the parameters of the fine-grained video LLM are in the middle instead of the highest sampling frames or granularity. This can be attributed to the variable length of video grounding segments sent from the coarse-grained video LLM, which requires the fine-grained video LLM to process both short videos and relatively longer videos (from 10 seconds to a few minutes).

Table 5: Hyperparameter experiment on the coarse-grained video LLM. The configs $(N, L)$ of fine-grained video LLM remain $(32, 12 \times 12 = 144)$.

| Hyperparameters | VideoMME | | | |
|---|---|---|---|---|
| $(N, L)$ | Short | Medium | Long | Overall |
| Duration(min) | $\leq 2$ | $4\sim 15$ | $30\sim 60$ | $1\sim 60$ |
| $(72, 8 \times 8 = 64)$ | **70.3** | **59.7** | 45.8 | 58.6 |
| $(288, 4 \times 4 = 16)$ | 67.7 | 57.8 | 51.4 | **58.9** |
| $(1152, 2 \times 2 = 4)$ | 66.3 | 55.0 | **52.2** | 57.8 |

Table 6: Hyperparameter experiment on the fine-grained video LLM. The configs $(N, L)$ of coarse-grained video LLM remain $(288, 4\times 4 = 16)$.

| Hyperparameters | VideoMME | | | |
|---|---|---|---|---|
| $(N, L)$ | Short | Medium | Long | Overall |
| Duration(min) | $\leq 2$ | $4\sim 15$ | $30\sim 60$ | $1\sim 60$ |
| $(8, 24 \times 24 = 512)$ | 66.4 | 56.2 | 48.6 | 57.1 |
| $(32, 12 \times 12 = 144)$ | **67.7** | **57.8** | 51.4 | **58.9** |
| $(128, 6 \times 6 = 36)$ | 67.5 | 57.3 | **51.5** | 58.8 |
| $(288, 4 \times 4 = 16)$ | 66.1 | 56.3 | 50.2 | 57.5 |

## A.3 ANALYSIS OF THE INFERENCE LATENCY AND REFLECTION CYCLES

Considering the inference latency and reflection cycles for the real-time application scenarios, as shown in Table 7, we compared the inference speed of our method and other Video LLM methods under the same settings using two A100-40GB GPUs. To avoid unlimited reflection iterations, we set a specific reflection threshold, which will terminate the reflection and select the answer response from the last round of video LLM as output if the iterated calls of two modules exceed a certain limit. By setting different thresholds of fine-to-coarse reflection cycles of our method, the inference speed and performance would vary. We could conclude that the current inference latency is understandable and consistent with the CrossVLLM framework since our method normally requires at least 2 separate inferences for coarse-grained and fine-grained Video LLMs. However, our method maintains a similar level of inference speed to existing Video LLMs, and considering the performance improvement brought by our method, this is acceptable for current long video understanding tasks.

We also report the number of reflection cycles required for our experiments, as shown in Table 8. The benchmarks for long video understanding tasks, on average, require more reflection cycles under our cross-granularity framework, and temporal video grounding tasks require less. This is because the long video understanding tasks are more complex and may require more iterations of key video segment grounding and fine-grained perception. Meanwhile, we can see that in the vast majority of cases from different benchmark datasets, the number of our reflection cycles is controlled within 1, which prevents expensive computational overhead caused by massive reflection iterations.

Table 7: Performance and inference speed comparison of our CrossVLLM methods with different thresholds of reflection cycles. The performance of Baseline Video LLMs represents their best performance of all the baseline models.

| Models | Inference Speed | Reflection Cycles | VideoMME | | | |
|---|---|---|---|---|---|---|
| | | | Short | Medium | Long | Overall |
| Baseline Video LLMs | 0.95~1.5 qa pairs/s | ✗ | 67.4 | 60.7 | 59.5 | 61.0 |
| Ours-CrossVLLM | 0.29~0.64 qa pairs/s | $\leq 3$ | 70.9 | **64.4** | **61.9** | **65.7** |
| | 0.45~0.64 qa pairs/s | $\leq 1$ | 70.9 | 64.2 | 61.4 | 65.5 |
| | 0.57~0.64 qa pairs/s | 0 | **71.8** | 60.3 | 53.8 | 62.0 |

Table 8: Ratio of reflection cycles required for different video understanding benchmarks. The threshold of reflection cycles is set to be 3.

| Datasets | Reflection Cycles | | | |
|---|---|---|---|---|
| | 0 | 1 | 2 | 3 |
| VideoMME | 80.1% | 17.5% | 1.8% | 0.6% |
| Lvbench | 73.9% | 21.4% | 3.2% | 1.5% |
| MLVU | 75.5% | 20.9% | 2.9% | 0.7% |
| ActivityNet Captions | 89.6% | 8.7% | 1.6% | 0.1% |
| Charades-STA | 91.2% | 7.6% | 1.2% | <0.1% |

## A.4 VISUALIZATION EXAMPLES

Here we present some inference examples of our CrossVLLM framework. In the examples, we omit the prompts, and policies' unimportant parts and the repeated questions to make them more concise. Figure 6 represents the examples of cross-granularity inference without the execution of fine-to-coarse reflection since the video segment to the question is relatively easy for localization, and the answer could be inferred through fine-grained perception at once. For example, in Figure 6(a), the video is a PPT animation with numerical titles (from 1 to 10), and the grounding condition is 'eighth point' that guides the coarse-grained video LLM to locate the key segment in the middle to back position of the video. Figure 6(b) is even more obvious as the condition 'finished' directly guides the coarse-grained video LLM to focus on the final part of the input long video.

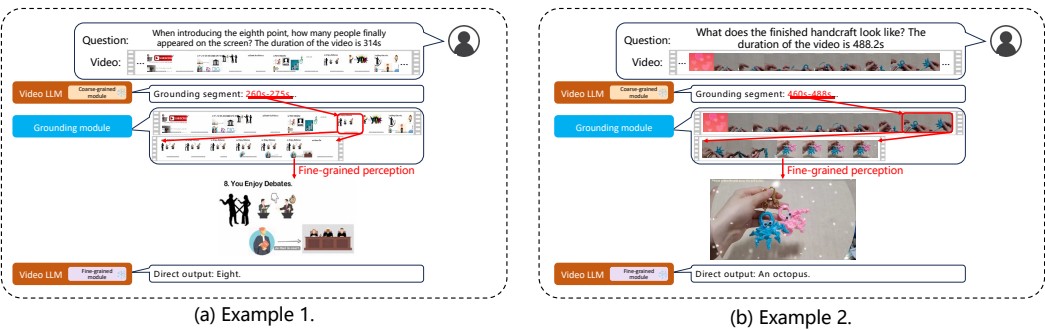

Figure 6: Visualization examples of cross-granularity inference without fine-to-coarse reflection.

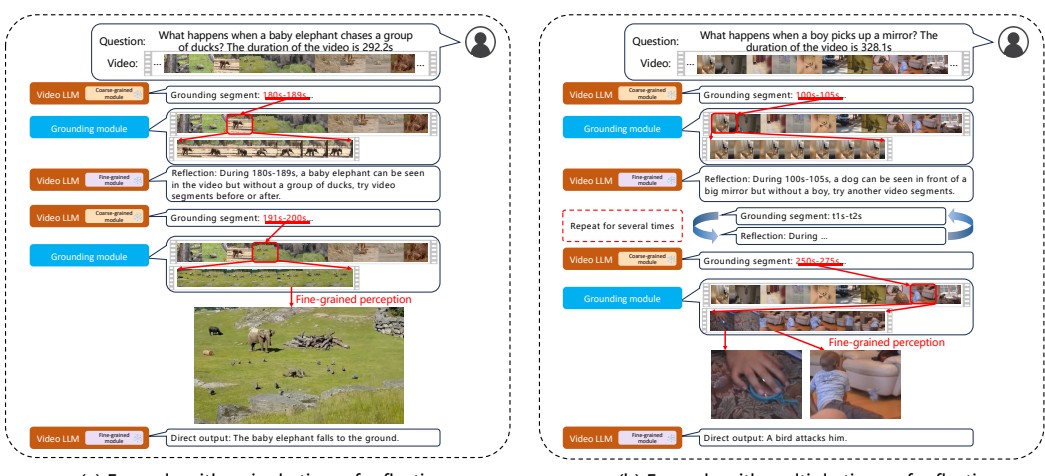

Figure 7: Visualization examples of cross-granularity inference. (a) represents an example with fine-to-coarse reflection once, and (b) represents an example with fine-to-coarse reflection repeated multiple times.

In addition, we provide examples of reflective granularity inference with reflection at least once shown in Figure 7. These long video understanding tasks are designed for the interaction of multiple objects, and they are original failure cases for the cross-granularity strategy without reflection. Figure 7(a) represents a video question example that requires a single time of fine-to-coarse reflection, and the regenerated grounding policy locates the correct key video segment for the second time. In the example of Figure 7(a), the grounding condition is *'a baby elephant chases a group of ducks'* and the video includes several segments distributed in different times, but all contain *'elephant'*, which interferes with the Video LLM to localize the correct video segment. Figure 7(b) represents

a video question example that is more difficult (as its condition is that *'a boy picks up a mirror'*, but *'mirror'* occurs across the entire video), and requires multiple fine-to-coarse reflections to finally locate the corresponding key video segment that fulfills the question entirely.

## A.5 LIMITATION AND FUTURE WORK

We discover some failure cases under the cross-granularity framework to further study how LLM analyzes long videos, and an example is provided in Figure 8. During this failure case, the coarse-grained video LLM is asked, *'How many magic shows are included in this video?'*, and the video LLM processes the long video input and responds with a grounded subset video segment that contains several magic shows and sends it to the fine-grained video LLM. However, since the grounded subset video contains *'magic shows'* content, related to the original video question, the fine-grained video LLM would confirm the grounded policy is effective and directly give the answer only based on the subset grounded video, ignoring that there might be other *'magic shows'* in the video segments that are not grounded.

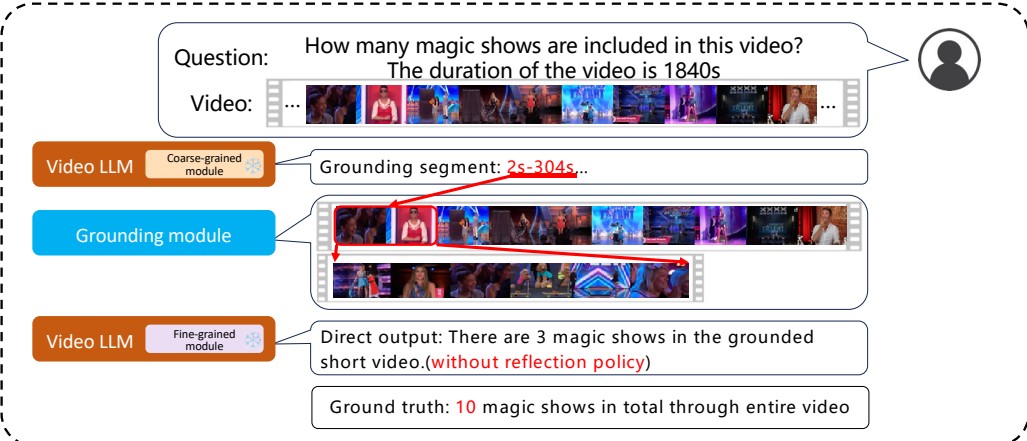

Figure 8: Example of a failure case. The coarse-grained video LLM accurately localized a subset video contains *'magic shows'* content related to the original video question. And then the fine-grained video LLM would confirm the grounded policy is effective and directly give the answer (without reflection policy) only based on the subset grounded video, ignoring that there might be other *'magic shows'* in the video segments that are not grounded.

This failure case originates from the insufficient prompting and policy design during our fine-to-coarse reflection. The optimal solution in such event-counting tasks is to ground and reflect segment by segment until the entire video is fine-grained perceived and summarize the count result, which calls for different prompting and policy design. Similar failure cases also occurred in action/event ordering tasks of long video understanding, and these would be the future direction of our method to further improve fine-grained long video understanding capabilities.

## B    Ethics Statement

This work adheres to the ICLR Code of Ethics. In this study, no human subjects or animal experimentation are involved. All datasets used, including ActivityNet-QA, Ego-QA, Next-QA, ActivityNet-Captions, Charades-STA, VTG-IT, VideoMME, Lvbench, and MLVU, are sourced in compliance with relevant usage guidelines, ensuring no violation of privacy. We have taken care to avoid any biases or discriminatory outcomes in our research process. No personally identifiable information is used, and no experiments are conducted that could raise privacy or security concerns. We are committed to maintaining transparency and integrity throughout the research process.

## C    Reproducibility Statement

We have made every effort to ensure that the results presented in this paper are reproducible. All datasets are publicly available to facilitate replication and verification. Our experimental setup, including training steps, model configurations, proposed algorithm, and hardware details, is described in detail in the paper. We have also provided a full description of **CrossVLLM** to assist others in reproducing our experiments.

Additionally, our experimental benchmarks for long video understanding and temporal video grounding, such as VideoMME, Lvbench, MLVU, Charades-STA, and ActivityNet-Captions, are publicly available, ensuring consistent and reproducible evaluation results. We believe these measures will enable other researchers to reproduce our work and further advance the field.

## D    LLM Usage

Large Language Models (LLMs) are not used to aid in the writing or polishing of our manuscript. It is important to note that the LLM is also not involved in our ideation, research methodology, or experimental design. All research concepts, ideas, and analyses are developed and conducted by the authors. The authors take full responsibility for the content of the manuscript.

