# OpenReview forum: "Cross-granularity Video LLM for Long Video Understanding"
_ICLR.cc/2026/Conference — ICLR 2026 Conference Withdrawn Submission_

### Official Review · Reviewer_xtTm · 2025-10-26

**Soundness:** 2
**Presentation:** 2
**Contribution:** 2
**Rating:** 2
**Confidence:** 5

**Summary:**

This paper proposes CrossVLLM, a cross-granularity Video LLM framework for long video understanding. The core idea is to use two adaptive modules with different token granularities to collaboratively handle long video inputs, including (1) a coarse-grained module for global temporal grounding and (2) a fine-grained module for detailed segment reasoning. The authors further design a coarse-to-fine grounding strategy and a fine-to-coarse reflection mechanism to iteratively refine the relevant video segments. In addition, a cross-granularity reinforcement learning strategy is introduced to optimize grounding performance. Experiments have been conducted on long video QA benchmarks (e.g., VideoMME, LVBench, MLVU) and temporal video grounding datasets (e.g., ActivityNet Captions, Charades-STA).

**Strengths:**

1. Overall, the cross-granularity idea is intuitive and reasonable, aligning with human perception and cognition processes.
2. The experiments were conducted on a wide range of tasks and benchmarks.

**Weaknesses:**

1. My major concerns are the novelty of the framework and the significance of the experiments. In the introduction and related work sections, the authors mentioned that existing methods "generally adopt uni-granularity tokens to represent the videos and then adopt the token pruning methods or frame sampling methods to reduce the token number", however, it should not be overlooked that a considerable number of works have already investigated the new paradigm of **coarse-grained grounding + fine-grained QA** for long video QA [1-5]. These important prior works seem not to be mentioned in the paper. This raises my concern regarding the novelty of the proposed framework.
2. What does "the test set of ActivityNet Captions" mean? I believe this dataset does not have a test set but only train, val_1, and val_2. The authors should clarify this.
3. The experiments also oddly omit some strong baselines without explanation. Some comparisons are **unfair** as well. For example, according to Page 14 *Module Training details*, the proposed model was trained on both Charades-STA and ActivityNet Captions. While in Table 2, the authors compare it with **zero-shot** baselines. Some other recent strong LLM-based VTG methods seem to be selectively missing. To name a few: VideoMind [5], TimeMarker [6], LLaVA-ST [7], E.T. Chat [8], and VideoExpert [9].
4. The baseline LLaVA-NeXT-Video-7B's performance is missing in the tables.

Minor Issue:
1. The authors did not claim to open-source their work (after acceptance), raising some reproducibility concerns given the limited implementation details in the paper.

[1] Self-Chained Image-Language Model for Video Localization and Question Answering
[2] Task Preference Optimization: Improving Multimodal Large Language Models with Vision Task Alignment
[3] A Simple LLM Framework for Long-Range Video Question-Answering
[4] Streaming Long Video Understanding with Large Language Models
[5] VideoMind: A Chain-of-LoRA Agent for Long Video Reasoning
[6] TimeMarker: A Versatile Video-LLM for Long and Short Video Understanding with Superior Temporal Localization Ability
[7] LLaVA-ST: A Multimodal Large Language Model for Fine-Grained Spatial-Temporal Understanding
[8] E.T. Bench: Towards Open-Ended Event-Level Video-Language Understanding
[9] VideoExpert: Augmented LLM for Temporal-Sensitive Video Understanding

**Questions:**

Please refer to the weakness section for my questions. My major concerns are about the novelty issue and unfair & not extensice comparisons in the paper. Strong justifications should be provided to demonstrate the significance of this work.

---

### Official Review · Reviewer_tULm · 2025-10-29

**Soundness:** 3
**Presentation:** 3
**Contribution:** 2
**Rating:** 4
**Confidence:** 5

**Summary:**

This paper explores a core challenge in video LLM. It proposes CrossVLLM with cross granularities for video understanding. The coarse-to-fine grounding helps localizes related temporal segments, and the fine-to-coarse reflection self-validates the correctness of grounded splits. The experiments on long video and temporal grounding benchmarks show the effectiveness.

**Strengths:**

1. The explored problem is meaningful and the motivation is clear. The idea of coarse-to-fine grounding and fine-to-coarse refection builds a rigorous inference trajectory for long video understanding.
2. The experiments are thorough. The extensive results on various benchmarks and the ablation studies validate the superiority of the proposed strategy.

**Weaknesses:**

1. The training data heavily relies on existing temporal grounding data, which are narrow in domain and restricts the generalization to open world videos.
2. The grounding module only considers single segment grounding and cannot deal with multi-hop scenarios.
3. The optimal number of the grounding and refection cycle varies with inference data. The authors only show results on general video data and more analysis on complex long videos that require strong temporal reasoning would be beneficial.

**Questions:**

The existing grounding data is very suitable to construct the training set for the grounding and reflection training, but the data diversity is quite limited. Is there any other ways to scale data without relying on too much priors?

---

### Official Review · Reviewer_gXAN · 2025-10-31

**Soundness:** 2
**Presentation:** 3
**Contribution:** 2
**Rating:** 4
**Confidence:** 3

**Summary:**

The paper proposes CrossVLLM, a two-module framework for long-video QA and temporal grounding: a coarse-grained video-LLM processes frames with fewer tokens per frame to locate salient segments, then a fine-grained video-LLM examines fewer frames with increased tokens to answer or refine timestamps. A fine-to-coarse reflection loop can send the pipeline back to re-ground if the segment shifts away. The training scheme mixes reinforcement learning with supervised fine-tuning.

**Strengths:**

1. Clear, modular recipe that can be implemented on top of LLaVA-Next-Video; the two-stage locate-then-inspect pipeline is sensible for long videos.

2. Training details are concrete: coarse/fine token budgets and N/L settings are explicit ((N, L) = (288, 16) vs. (32, 144) at inference), and the SWIFT-based alternation between DPO and SFT is described.

3. Inference control: the authors quantify reflection cycles; the writing is clear and the method is well presented.

**Weaknesses:**

1. Lack of supervision for reflection and low trigger rate.

2. The reflection behavior is neither supervised nor weakly supervised, leading to no reflection on ~80% of samples across many datasets. For ~30-minute videos, it is unrealistic to achieve consistently precise, single-shot, second-level localization on most examples. This undermines the effectiveness of the proposed coarse-localization + fine-reflection design for long videos. Table 8 shows this across five video benchmarks. Moreover, per Table 2, the method does not achieve competitive performance against many recent agent-style long-video frameworks that rely on back-and-forth deliberation or reflection. Together, these findings suggest that without supervision, multi-round reflection largely fails on long videos, primarily because the current approach cannot reliably trigger reflection.

3. Novelty claim (“first cross-granularity video-LLM”) is overstated.

4. Coarse temporal localization followed by reflection-based correction is already common in long-video understanding, and this work does not introduce clear innovations in either component. In practice it fine-tunes localization with training data and then uses an LLM reflection prompt to textually revise the previous localization. The more serious concern is that, in most cases, reflection is not triggered.

5. Missing baselines/SOTA and protocol transparency.

6. Key temporal grounding models and recent long-video baselines are absent (e.g., TimeChat, Video-r1, VideoChat-R1.5 with iterative perception and RL/test-time scaling, Video-XL2 for hour-scale long-video VLMs). Without these, SOTA claims are weak. In addition, on Video-MME/LVBench/MLVU it is unclear whether subtitles/audio were disabled; this affects comparability and fairness and can inflate scores (e.g., LVBench/MLVU emphasize long-horizon reasoning, and community practice varies).

**Questions:**

1. On “firstness”: How does CrossVLLM meaningfully differ from prior coarse→fine + reflection pipelines (e.g., VideoAgent/VideoTree)?

2. Lines 210–212 state: “Since video LLM with the coarse-grained module may encounter relatively simple video questions for short videos, it is also prompted [and] allowed to give policies with direct output answers in simple video cases.”

3. Could this design confuse reinforcement-learning training by mixing “direct answering” policies with “localize-then-answer” policies?

4. For each iteration, are the localization boxes produced against the entire video using the latest coordinates, or by some other mechanism? If it is the former, how do you ensure that reflection produces different grounding each time? Do you observe duplicate grounding IDs or near-duplicates? If so, how does the method address this?

5. (See Weakness 1.) Please report, on VideoMME-Long and LVBench, the number of reflection rounds and the corresponding accuracy. This will help clarify the method’s behavior.

6. Modality control: Please clarify subtitle/audio usage on Video-MME and report with/without results to ensure fair comparison (community toggles vary).

---

### Official Review · Reviewer_vEXC · 2025-10-31

**Soundness:** 3
**Presentation:** 2
**Contribution:** 3
**Rating:** 6
**Confidence:** 4

**Summary:**

This paper introduces CrossVLLM, a novel VideoLLM framework designed to address the challenges of understanding long videos. CrossVLLM leverages adaptive modules operating at different granularities to improve both high-level semantic and fine-grained visual understanding. It achieves state-of-the-art results on challenging long video tasks.

**Strengths:**

1. The core idea is technically solid and makes sense. The model demonstrates strong performance, especially using the coarse-to-fine retrieval and fine-to-coarse reflection, boosting the performance compared to one-round inference.


2. The ablation study in Table 3 is insightful, effectively demonstrating the contribution of each component, particularly the impact of RL tuning.



3. The training implementation is described in detail, which lowers the barrier for reproducibility within the community.

**Weaknesses:**

1. The current design of Figure 1(b) has some flaws, particularly with the arrows, which could be polished for better clarity and visual flow.
Figure 1(c) and Figure 2 appear very similar, which may cause confusion for readers. Additionally, the interpretation of Figure 2 is unclear—specifically, the meaning of "Output" after the "Answer Policy" step is ambiguous.

2. The benchmarking results in Table 1 are missing strong baselines, such as Qwen2.5-VL and Video-XL 2.

**Questions:**

I do not have specific questions, please see my weakness.

---

### Note · Authors · 2025-11-13

**Comment:**

We have read all the suggestions and concerns given by reviewers. After careful consideration, we have decided to withdraw our submission. We would like to sincerely thank all the reviewers for taking the time to review our paper and provide insightful feedback and suggestions.

**Withdrawal Confirmation:**

I have read and agree with the venue's withdrawal policy on behalf of myself and my co-authors.